# Comparison of Conventional and Individualized 1-MET Values for Expressing Maximum Aerobic Metabolic Rate and Habitual Activity Related Energy Expenditure

**DOI:** 10.3390/nu11020458

**Published:** 2019-02-22

**Authors:** Juliane Heydenreich, Yves Schutz, Katarina Melzer, Bengt Kayser

**Affiliations:** 1Swiss Federal Institute of Sport Magglingen, 2532 Magglingen, Switzerland; juliane.heydenreich@gmail.com (J.H.); katarinamelzer@hotmail.com (K.M.); 2Institute of Sport Sciences, University of Lausanne, 1015 Lausanne, Switzerland; 3Department of Physiology, University of Fribourg, 1700 Fribourg, Switzerland; yves.schutz@unifr.ch

**Keywords:** resting metabolic rate, maximum oxygen consumption, energy expenditure, endurance athletes

## Abstract

The maximum aerobic metabolic rate can be expressed in multiple metabolically equivalent tasks (MET), i.e., METmax. The purpose was to quantify the error when the conventional (3.5 mL∙kg^−1^∙min^−1^) compared to an individualized 1-MET-value is used for calculating METmax and estimating activity energy expenditure (AEE) in endurance-trained athletes (END) and active healthy controls (CON). The resting metabolic rate (RMR, indirect calorimetry) and aerobic metabolic capacity (spiroergometry) were assessed in 52 END (46% male, 27.9 ± 5.7 years) and 53 CON (45% male, 27.3 ± 4.6 years). METmax was calculated as the ratio of VO_2_max over VO_2_ during RMR (METmax_ind), and VO_2_max over the conventional 1-MET-value (METmax_fix). AEE was estimated by multiplying published MET values with the individual and conventional 1-MET-values. Dependent *t*-tests were used to compare the different modes for calculating METmax and AEE (α = 0.05). In women and men CON, men END METmax_fix was significantly higher than METmax_ind (*p* < 0.01), whereas, in women END, no difference was found (*p* > 0.05). The conventional 1-MET-value significantly underestimated AEE in men and women CON, and men END (*p* < 0.05), but not in women END (*p* > 0.05). The conventional 1-MET-value appears inappropriate for determining the aerobic metabolic capacity and AEE in active and endurance-trained persons.

## 1. Introduction

The aerobic capacity (maximum oxygen consumption, VO_2_max) is defined as the highest rate at which oxygen can be taken up and utilized by the body during an intense large muscle group exercise [1]. It is used in both athletic and health settings, as a determinant of physical performance [1] or as a predictor of health risk and longevity [2]. Traditionally, VO_2_max is expressed as the ratio of maximum rate of oxygen consumption and body mass (mL∙kg^−1^∙min^−1^). However, expressing VO_2_max by normalizing for body mass (*m*) can be problematic since an *m*-based ratio is negatively correlated with *m* [3] and, therefore, imposes a penalty in heavier subjects, especially since the actual scaling with *m* is not linear [4]. This ratio is, thus, inappropriate for studies where VO_2_max is compared between groups that are not matched for body size and mass, or when body mass changes over time [5,6]. One way to remove the effects of *m* is to adjust VO_2_max by using the power function relationship VO_2_max = *am*^k^ where *a* is the scaling constant and *k* is the scaling exponent [5]. However, there is considerable controversy regarding the theoretical value this exponent should take (e.g., *k* = 2/3, 3/4 or >3/4) [7,8]. In addition, the effect of *m* is also a function of body composition since muscle volume is an important determinant of metabolic capacity while fat tissue is comparatively metabolically inert. This would imply that fat mass changes would introduce a greater bias as compared to lean mass changes.

Alternatively, the aerobic capacity can be expressed as the maximum aerobic metabolic rate in a multiple of metabolic equivalent of tasks (MET), i.e., METmax. One MET is defined as the energy expended by a subject at rest (resting metabolic rate, RMR) of ~1 kcal·kg^−1^·h^−1^ [9], and is equivalent to a volume of oxygen consumed of 3.5 mL O_2_ kg^−1^·min^−1^ [10]. The MET provides a useful way to describe and classify physical activities by expressing the specific level of activity energy expenditure (AEE) (under steady state conditions) in relative value, i.e., as a multiple of RMR. Theoretically, 10 METs would then correspond to 35 mL O_2_ kg^−1^ ·min^−1^, which is equivalent to ~10 kcal·kg^−1^·h^−1^.

The Compendium of Physical Activities provides a five-digit coding scheme linking categories and types of physical activity with their respective intensity values in METs [9]. It was originally developed for use in epidemiologic and surveillance studies to standardize the MET intensities for various types of physical activity used in questionnaires. However, the Compendium is also frequently applied for determining precise energy costs of activities outside of its original scope. In several studies, where physical activity questionnaires were applied, the energy expenditure was estimated by using established MET codes from the Compendium of Physical Activity [11,12,13,14,15,16].

Several authors have questioned the widespread application of the conventional 1-MET value [17,18,19,20]. The value was derived from measurements of resting oxygen consumption (resting metabolic rate, RMR) of just one person, who is a 70-kg, 40-year old male, and it was shown that this value over-estimates [19,20,21,22,23,24,25,26] or underestimates [19] RMR for many types of individuals. RMR is lower in overweight subjects, declines with age, and is lower in females compared to males [20]. Therefore, estimation of AEE using the conventional 1-MET value might misrepresent actual energy expenditure. This might lead to inaccurately estimating energy requirements resulting in a positive or negative energy balance and undesirable and unexpected weight fluctuations. In addition, the maximum aerobic metabolic rate expressed as METmax might be erroneous when the conventional 1-MET value is used instead of the actual metabolic rate at rest. However, the correct assessment of oxygen consumption at rest (resting metabolic rate; RMR) requires considerable expense for both participants and researchers. Therefore, several prediction equations were developed to estimate RMR (e.g., Harris-Benedict [27], Cunningham [28]), but these also have their limitations [29,30].

Purpose of the study was to quantify the error when the conventional compared to an individualized 1-MET value is used for (1) calculating METmax, and (2) estimating energy expenditure for various daily physical activities, in endurance trained women and men, and active healthy controls. It was further investigated whether the use of a predicted RMR by the Harris-Benedict equation would reduce such an error. It was hypothesized that the use of the conventional 1-MET value would lead to relevant errors in both calculating METmax and estimating energy expenditure in comparison with an individualized approach.

## 2. Materials and Methods

### 2.1. Participants

After a public announcement, 68 competitive endurance athletes (31 women, 37 men; regular endurance training volume ≥300 min∙wk^−1^ and participation in competitions) and 63 healthy, non-endurance-trained active controls (34 women, 29 men; max. 150 min∙wk^−1^ moderate endurance training) were recruited for this study. Inclusion criteria for all participants were weight stability (<2 kg of weight difference in the last 3 months), a Body Mass Index (BMI) between 18.5 and 25 kg∙m^−2^, and an age between 18 and 40 years. All participants were non-smokers, not pregnant or lactating, not dieting, not suffering from metabolic disease and/or eating disorders, and not taking medication (apart from contraceptives). Athletes did not change their training habits within the last four weeks before the experiments (e.g., due to injury or disease).

The Regional Ethics Committee of the Canton Berne, Switzerland (KEK-number 090/15) approved all experimental procedures, and the study was carried out according to the recommendations of the latest Helsinki Declaration. Written informed consent of the participants was obtained before any testing.

### 2.2. Study Design

The participants arrived to the testing center on two separate testing days and had refrained from strenuous physical activity for at least 24 h. On the first testing day, the participants completed, in a fasting state (≥12 h absence of any food or fluid intake, ≥36 h absence of alcohol or caffeine intake) measurements in the following order: (1) anthropometry and body composition, (2) RMR, and (3) individual calibration of a combined heart rate (HR) and movement sensor (see below). One week after the first testing day, the participants performed an incremental exercise test (VO_2_max) in a non-fasted state. On the days between the two testing days, the participants wore the HR and movement sensor for at least 7 days. All tests were carried out in Magglingen (Switzerland) at an altitude of 950 m.

### 2.3. Anthropometric Data and Body Composition

Height and body mass were measured to the nearest 0.5 cm and 0.1 kg using a height rod (Seca 213, Seca, Hamburg, Germany) and a calibrated beam scale (Seca 877, Seca, Hamburg, Germany), respectively, with the participants in light clothing and without shoes.

Body composition was assessed using Lunar iDXA (GE Healthcare, Madison, WI, USA). The iDXA was calibrated on a daily basis using a calibration phantom before any testing. The participants were in underwear, bladder-voided, and all metal artefacts were removed. During the measurement, participants were in a supine position on the scanning table with their ankles and legs fixed using supports. Arms were positioned to the side with the palms flat on the table. Participants were requested not to move during the measurement. Whole body scans were performed, according to the manufacturer’s instructions. Adipose tissue mass, lean tissue mass, and bone mineral content were derived with the accompanying software (enCore software v. 11.10, GE Healthcare, Madison, WI, USA). Estimation of total body composition with the Lunar iDXA has been reported to be excellent in other studies [31,32].

### 2.4. Resting Metabolic Rate

Following the body composition assessment, RMR was measured by indirect calorimetry using a ventilated hood system (Quark CPET, COSMED Srl, Rome, Italy). Calibrations of the flowmeter and gas analyzer were carried out before each test, according to the manufacturer’s instructions. Participants were acclimatizing and relaxing for 30 min on a bed before the hood was placed over the participant’s head and measurements were started. VO_2_ and carbon dioxide production (VCO_2_) were measured for 30 min at 10-s intervals with participants remaining motionless in a supine position in a thermo-neutral environment (20–25 °C [33]). The first 5 min were eliminated as the acclimatization artifact. From the remaining 25 min, the interval of 5 consecutive minutes with the lowest means of the coefficients of variation (CV) for VO_2_ and VCO_2_ was chosen. By use of the abbreviated Weir equation, RMR was calculated [34]. Pre-hoc exclusion criteria were values of CV of VO_2_ and VCO_2_ ≥10% and respiratory quotient (RQ, defined as the ratio of VCO_2_ and VO_2_) <0.70 and >1.00, since values outside the plausible range for RQ suggest protocol violations or inaccurate gas measurements [33]. Average RQ during RMR measurement was 0.76 ± 0.04 and the CV of VO_2_ and VCO_2_ were 3.6 ± 1.5% and 4.5 ± 1.9%, respectively. RMR measurements took place at a mean temperature of 21.9 ± 1.1 °C, 39.1 ± 10.3% humidity, and an air pressure of 914 ± 8 hpa.

### 2.5. Measurement of VO_2_max

Before the test, each participant filled out the German [35] or French [36] version of the Physical Activity Readiness Questionnaire (PAR-Q). Only if participants answered all items with “no,” the exercise testing was started. The test was performed on a treadmill (women: model Mercury, men: model Venus, h/p/Cosmos Sports & Medical GmbH, Traunstein, Germany). After a 5-min warm-up jog, non-athletic participants began running at 7 km∙h^−1^, whereas participants from the athletic group started at 9 km∙h^−1^. The speed was increased by 1 km∙h^−1^ every minute for the first 3 min of the test, and, thereafter, by 0.5 km∙h^−1^ every 30 s until exhaustion. Treadmill inclines were set at 4° throughout the test [37]. Gas exchange was measured breath-by-breath with an open-circuit system (Quark CPET, COSMED Srl, Rome, Italy). Calibration was performed before each test, according to the manufacturer’s instructions. VO_2_ data was processed using 10-s time averages and VO_2_max was determined as the highest 30-s VO_2_ average for the test [38]. HR was continuously registered with a wireless HR monitoring system (model SZ990, COSMED Srl, Rome, Italy). The participants’ rating of perceived exertion (RPE) was assessed immediately after the test with Borg’s RPE scale [39]. If the primary criteria of a plateau in oxygen uptake (defined as an increase of VO_2_ <2.1 mL∙kg^−1^∙min^−1^ [40]) was not reached by the participant (*n* = 4), then the secondary criteria of a RQ value ≥1.10, and an HR close (± 10 bpm) to the age-predicted maximum HR [41] were used to determine whether the participant reached maximal effort and VO_2_max [42]. VO_2_max-tests were carried out at a mean room temperature of 21.7 ± 1.2 °C, a humidity of 39.2 ± 9.8%, and an air pressure of 914 ± 7 hpa. In general, a temperature range of 20 to 22 °C in a cool, dry environment (<50% humidity) is considered comfortable for exercise testing [43].

### 2.6. Calculation of METmax and Estimation of Energy Expenditure

METmax was calculated in two modes, which are the ratio of (1) VO_2_max (mL∙min^−1^) over VO_2_ during RMR measurement (METmax_ind), and (2) the VO_2_max (mL∙min^−1^) over the conventional 1-MET value (3.5 mL∙kg^−1^∙min^−1^, METmax_fix). For calculating activity energy expenditure (AEE) activities, different MET intensities were chosen: light (<3 METs), moderate (3–5.99 METs), vigorous (6–8.99 METs), and very vigorous (≥9 METs) activity [44]. MET values for the different activities were achieved by using the Compendium of Physical Activities [9]. AEE of the different activities was estimated by multiplying the MET value (1) with the individual 1-MET value (AEE_ind), (2) with the conventional 1-MET value (AEE_fix), and with the predicted RMR by using the Harris-Benedict equation (AEE_pred). For estimating the total energy expenditure (TEE), three different physical activity levels (PAL) were chosen including a sedentary or light activity lifestyle (PAL 1.53), a moderately active or active lifestyle (PAL 1.76), and a vigorously active lifestyle (PAL 2.25) [45]. The different PAL values were then multiplied (1) with the individual 1-MET value (TEE_ind), (2) with the conventional 1-MET value (TEE_fix), and (3) with the predicted RMR by use of the Harris-Benedict equation (TEE_pred).

### 2.7. Physical Activity Level (PAL)

The PAL of the participants was assessed using a combined HR and movement sensor (Actiheart; Cambridge Neurotechnology Ltd., Papworth, UK). The Actiheart was clipped onto two standard ECG electrodes (3M^TM^ Red Dot^TM^ Electrode 2560; 3M Health Care, St. Paul, USA) on the chest of the participant, according to the manufacturer’s instructions, and worn day and night [46]. The device was calibrated for each participant using a standard step test, which is a built-in function of the Actiheart software version 4.0.92 (Cambridge Neurotechnology Ltd., Papworth, UK). AEE was estimated by analyzing 6 full-day (24 h) recordings of HR and body movement with a 15-s averaging epoch setting. Participants were requested to continue their habitual life routine and physical activities during the recording period. TEE was calculated as the sum of RMR, AEE, and diet-induced thermogenesis (estimated as 10% of TEE [47]). PAL was then calculated as TEE/RMR. The Actiheart was shown to give accurate estimations of AEE during a wide range of activities in male and female subjects of various ages, body mass, and fitness levels [48,49,50,51,52].

### 2.8. Statistics

Statistical analyses were performed with SPSS statistics version 24 for MS-Windows (IBM Corp., Chicago, IL, USA). Mean values and standard deviations (SD) were calculated and data was checked for normality using the Shapiro-Wilk-test. All parameters were normally distributed with the exception of age, body mass, body mass index (BMI), body fat (%), fat-free mass (FFM; kg), RMR (kcal∙day^−1^), VO_2_max (L∙min^−1^), RQrest, and AEE/TEE calculated either by use of the conventional, predicted, or individual 1-MET value. Group differences were tested by independent *t*-tests and Mann-Whitney-U-tests (α = 0.05). The relationship between the two modes for calculating METmax and the relationship between VO_2_max and RMR were first investigated using the Pearson’s Product moment correlation analysis. The correlation coefficients (*r*) were classified, according to Cohen [53]. An *r* between 0.10–0.29 was considered small, between 0.30–0.49 was considered moderate, and between 0.50–1.0 was considered showing a strong association. The data were further analyzed using dependent *t*-tests. In addition, the mean absolute error (MAE) and the mean absolute percentage error (MAPE) of METmax_fix compared to METmax_ind were calculated. Since no standardized threshold exists for high or low MAPE, a MAPE ≥10% was considered an indicator of inaccuracy as suggested by other authors [54,55,56,57]. The standard error of the estimate (SEE) was calculated by linear regression, where METmax_ind was entered as a dependent variable and METmax_fix as an independent variable. For differences in estimating AEE/TEE by use of the individual, predicted, and conventional 1-MET value, the Wilcoxon signed-rank test for dependent samples was applied.

## 3. Results

### 3.1. Participants

Nine participants who did not meet the pre-defined inclusion criteria (e.g., BMI <18.5 or >25.0 kg∙m^−2^, age <18 or >40 years, not weight stable), 9 participants with invalid RMR tests (e.g., RQ <0.70 or >1.00, CV of VO_2_ and VCO_2_ >10%), and 7 subjects without a valid VO_2_max test (e.g., no plateau or other criteria for the maximal effort reached) were excluded from the analysis. One participant withdrew from the study due to personal reasons. In total, data of 57 women and 48 men were included in the analysis. The subjects were grouped, according to their aerobic fitness level (METmax_ind). Male and female participants with a METmax_ind above the 50th percentile were classified as endurance trained participants (END, *n* = 24 and *n* = 28, respectively). Subjects with a METmax_ind below the 50th percentile served as healthy, non-endurance trained active controls (CON, 24 men and 29 women). In Table 1, anthropometric data, body composition, RMR, and VO_2_max of the participants are displayed. Women END had a significantly lower body fat percentage and BMI and higher PAL than CON (*p* < 0.05). In men, no significant differences between groups were obtained for body composition, RMR, and PAL (*p* > 0.05). The individual 1-MET value was significantly higher than 1 kcal∙kg^−1^∙h^−1^ in men and women CON, and men END (*p* < 0.05). When RMR was predicted by the use of the Harris-Benedict equation, the RMR was significantly lower than 1 kcal∙kg^−1^∙h^−1^ in women CON, and higher in men CON and END (*p* < 0.05). The range of VO_2_max was 2.2–3.9 L∙min^−1^ or 34.4–62.0 mL∙kg^−1^∙min^−1^ for women and 2.5–5.3 L∙min^−1^ or 42.8–78.4 mL∙kg^−1^∙min^−1^ for men, respectively. Men and women END had significantly higher aerobic capacity compared to CON (*p* < 0.01). In Figure 1, the relationship between VO_2_max and RMR is displayed. There was a significant positive relationship between VO_2_max and RMR in all subgroups (*p* < 0.0001).

### 3.2. Calculation of METmax

METmax_ind and METmax_fix in women and men CON correlated (*r* = 0.69 and *r* = 0.78, respectively, *p* < 0.0001, Table 2). The MAE, MAPE, and SEE are presented for all groups in Table 2. In women and men CON, and men END METmax_fix significantly overestimated METmax (*p* < 0.01), whereas, in the women END, there was no difference (*p* > 0.05). When looking at the total sample, METmax_fix significantly overestimated METmax, compared to the use of the individual 1-MET value for its calculation (16.0 ± 2.2 vs. 15.1 ± 1.6, *p* < 0.0001). The range of MAPE was 6.6% to 11.3% across all groups. METmax_ind was significantly higher in men and women END compared to their non-athletic counterparts (*p* < 0.0001, Figure 2).

### 3.3. Estimation of AEE/TEE

The conventional 1-MET value significantly underestimated the energy expenditure of all activities in men and women CON and men END (*p* < 0.05), whereas, in women END, no difference was observed (*p* > 0.05, Table 3). For example, when the energy expenditure during one hour of running was estimated by use of the individual 1-MET value, the AEE was in the mean 32 kcal∙h^−1^, 97 kcal∙h^−1^, and 84 kcal∙h^−1^ higher in women and men CON, and men END, respectively, compared to the use of the conventional 1-MET value for its calculation. When AEE was calculated based on an RMR estimated by use of the Harris-Benedict equation in all subgroups, estimated AEE was significantly lower than AEE_ind for all activities (*p* < 0.05).

TEE was significantly underestimated in men and women CON, and men END when the conventional 1-MET value was used for estimating RMR and a PAL of 1.53, 1.76, or 2.25 was applied for calculating TEE (*p* < 0.05, Table 4). The range of the mean difference between TEE_ind and TEE_fix was 41–222 kcal∙day^−1^ for a PAL of 1.53, 47–255 kcal∙day^−1^ for a PAL of 1.76, and 60–326 kcal∙day^−1^ for a PAL of 2.25 across all groups. When TEE was calculated based on an RMR estimated by use of the Harris-Benedict equation, estimated TEE was still significantly lower than TEE_ind for all PAL values in all subgroups (*p* < 0.05).

## 4. Discussion

Aims of the study were to quantify the absolute and relative errors when the conventional 1-MET estimated value (defined as a constant viz. 1 kcal·kg^−1^·h^−1^) was compared to an individualized RMR value measured by indirect calorimetry. Both values were used as the baseline for calculating METmax and they were compared to determine whether the use of a measured (rather than predicted) RMR would reduce the relative and absolute error of prediction of METmax.

In endurance trained men of the present study, METmax was significantly overestimated and predicted that the resting energy expenditure was slightly underestimated when the conventional standard 1-MET value was used for their calculation. In the endurance trained women, no differences between the conventional vs. individual 1-MET value were found so that the estimation of METmax was marginally higher by 0.4 METs only, as compared to the measured value (Table 2).

In men and women controls, and endurance trained men, the individual 1-MET value was significantly higher than the conventional and fixed 1-MET value (*p* < 0.05). Therefore, it can be concluded that the use of the conventional 1-MET value is inappropriate for determining the aerobic metabolic capacity and estimating the daily activity related energy expenditure (using a METs table) in active people and endurance trained athletes.

These findings are in contrast to the majority of published studies, where the measured individual 1-MET value in women was mostly lower than 3.5 mL∙kg^−1^∙min^−1^ or 1 kcal∙kg^−1^∙h^−1^. For example, in a study of Byrne et al., the mean resting energy cost was 2.56 mL∙kg^−1∙^min^−1^ or 0.84 kcal∙kg^−1^∙h^−1^ [17]. However, they measured RMR in a large, heterogeneous sample, comprising many women and less men, with a wide age range (18–74 years) as well as the BMI range (13.8–57.5 kg∙m^−2^). Indirect calorimetry measurements were made (1) with a comfortable hood system (and not with face mask or mouthpiece, which are known to generate slightly higher RMR values, [58]), (2) under strictly standardized conditions with participants fasting for at least 12 h and no exercise allowed the day preceding testing, and (3) a 25-min period at steady state of a total 45 min RMR measurement was chosen for analysis. They also found that fat mass and FFM accounted for 62% of the variance in resting VO_2_. In a review of McMurray et al. examining RMR in healthy adults, the mean value for RMR was 0.86 kcal∙kg^−1^∙h^−1^, as expected, which is higher for men than women, decreases with increasing age, and is less in overweight/obese than normal weight adults [20]. Adults with a BMI ≥30 kg∙m^−2^ had the lowest RMR (<0.74 kcal·kg^−1^·h^−1^).

In a previous study, the RMR of adolescents, pregnant and post-pregnant women, and active men was measured [19]. A significantly higher relative RMR in adolescents compared to the conventional 1-MET value was found, whereas, in the other subgroups, no differences were observed. When reviewing data on endurance trained men and women whose RMR was measured using indirect calorimetry, similar results to the present study were obtained: women were expending, on average, 1.11 kcal·kg^−1^·h^−1^, and men 1.13 kcal·kg^−1^·h^−1^ [59,60,61,62,63,64,65,66,67]. In these experimental studies, the use of the conventional fixed 1-MET value led to considerably greater error in endurance trained men and women compared to the measured RMR value. In the present study, endurance trained men also demonstrated a significantly higher individual RMR value compared to the conventional theoretical 1-MET value. In addition, participants in the control groups also had significantly higher individual 1-MET values. Incidentally, it should be noted that the participants who were assigned to the control group are not representative of the general population of Switzerland, which comprises a majority of moderately active and sedentary individuals. In other words, the individuals of the present sample were more physically active and had a “normal” BMI. For comparison purposes, Swiss men with a mean age of 42 years had a relative body fat percentage of 21.0% [68], which is a slightly elevated value. In a sample of Swiss women (*n* = 64) with a mean age of 27 years, more than two-thirds (70.3%) had a body fat percentage ≥30% [69], which indicates the presence of a high percentage of plump women. Since fat mass is the strongest predictor of the variability of resting VO_2_ [17], such differences in body composition (higher relative FFM in the present study) might explain the higher individual 1-MET value of the controls compared to the values reported in the literature. In any case, these findings underline the limits of using a fixed standard 1-MET value.

### 4.1. Calculation of METmax

The present study addressed how much relative and absolute error the use of the conventional 1-MET value would introduce when used for calculating METmax. In 79% of the women and 88% of the men CON, and 83% of the endurance trained men (END), the conventional 1-MET value overestimated METmax, which resulted in a significant overestimation of the mean aerobic metabolic capacity in these groups. The MAPE was 6.6%, 10.3%, and 11.3% in both women and men CON, and men END, respectively, and 35%, 50%, and 46% of the women and men CON, and men END had a MAPE ≥10%, respectively. Generally, a MAPE ≥10% can be considered to be a marker for inaccurate measurements [54,55,56,57]. Therefore, the authors of the present study strongly encourage researchers and any other person, who wants to determine the aerobic metabolic capacity of active subjects, to measure RMR before a maximum exercise test is conducted. Since proper assessment of RMR requires further expertise, equipment, and time, and is somewhat cumbersome for the participant, RMR should be at least *estimated* using established equations, such as the Harris-Benedict [27] or the Cunningham [28] formulas. Another possibility would be the direct assessment of resting VO_2_ of the subject prior to exercise testing while standing still on the treadmill or sitting quietly on a bike. However, it is unclear whether this resting VO_2_ is more appropriate for calculating METmax than using a conventional or estimated value for RMR.

### 4.2. Estimation of AEE, TEE, and Physical Activity Level (PAL = TEE/RMR)

As the second purpose, the error when the conventional 1-MET value was used for estimating daily energy expenditures was investigated. Large differences using the individual and conventional 1-MET values for estimating energy expenditures were observed. In women and men CON, and men END, the conventional 1-MET value significantly underestimated the energy expenditure of several physical activities and total daily energy expenditure. For example, using a PAL of 2.25 for estimation of TEE, reflecting a rather active lifestyle, led to an underestimation of energy requirements of 108 kcal∙day^−1^, 326 kcal∙day^−1^, and 282 kcal∙day^−1^ for women and men CON, and men END, respectively, when the conventional 1-MET value of 1 kcal∙kg^−1^∙h^−1^ was used. In the endurance trained women of this study, no difference between the individual and conventional 1-MET value was found when energy requirements were calculated. This is because the difference between the individual and conventional 1-MET value was not significant.

PAL and MET values are frequently used for estimating energy requirements in athletes. However, using the conventional 1-MET value for their calculation might underestimate (true) energy costs of their activities and might promote insufficient energy intake, especially in situations when athletes wish to control their energy balance (e.g., for weight loss or maintenance). Besides the possible undesirable effects on body mass and body composition, the underestimation of energy costs and further advised erroneous energy intakes might also lead to a higher risk of suffering from Relative Energy Deficiency in Sport (RED-S), its concomitant symptoms, and a decrease in endurance performance [70]. Therefore, it can be recommended to either (1) measure directly the energy costs of physical activities or daily energy requirements using validated and objective measures, (2) measure RMR and use the individual 1-MET value for estimating energy requirements, or, in the case that both options are not possible, to (3) estimate RMR using established formulae and use a corrected 1-MET value for estimating the energy expenditure.

Most often in the general population, the individual 1-MET value is significantly lower than the conventional 1-MET value [19,20,21,22,23,24,25,26]. Therefore, in the general population, the use of the conventional 1-MET value is mostly *overestimating* energy costs of activities, as shown previously by others [21,23]. This overestimation of energy requirements might, thus, promote a positive energy balance and could contribute to a higher risk for obesity and concomitant diseases. Several authors recommend the use of corrected MET values to account for personal variation in sex, body mass, height, and age in order to estimate the individual physical activity level more accurately [17,18]. Hereby, the standard 1-MET value of 3.5 mL∙kg^−1^∙min^−1^ is divided by a predicted RMR obtained from the Harris-Benedict equation [27]. The authors found a significant reduction of underestimation and misclassification of the MET values when a corrected 1-MET value was used. Howley (2011) stated that the ratio of the work metabolic rate to measured RMR should not be called “METs,” since METs are, by definition, restricted to a denominator of 3.5 mL∙kg^−1^∙min^−1^ [71]. In the present study the use of a predicted 1-MET value by using the Harris-Benedict equation reduced the mean difference in energy expenditure estimation between the use of a measured and estimated RMR in men, whereas, in women, the mean difference was even higher than the use of the conventional 1-MET value. Therefore, the use of corrected METs might be useful for estimating individual energy costs in some cases, whereas the standard MET values can help classify the intensity of physical activities, when different studies are compared. Lastly, it must be stated that neither the standard nor the use of a corrected 1-MET value can replace the direct assessment of energy expenditure by either measuring oxygen consumption during physical activities or by using the doubly labeled water technique.

### 4.3. Strengths and Limitations

This is the first study with the purpose to assess the individual 1-MET value in endurance trained athletes. Expressing aerobic capacity as a ratio of maximum oxygen consumption divided by oxygen consumption at rest is a suitable measure in endurance trained athletes and healthy, active controls. A big advantage of the METmax calculation is that the denominator (RMR) already takes into account the metabolic and physiological characteristics of the individual at the baseline, so that inter-individual (e.g., comparisons between groups of different age, sex, body composition, physiological status, ethnicity) and intra-individual (e.g., change of body composition throughout different observation time points) comparisons of METmax will not be biased by a difference or change of these characteristics. The readjustment of RMR (upward or downward) will not bias the validity of the new METmax recalculation. Another advantage of the present study is the focus on strict protocols for assessment of the variables, e.g., RMR was measured in the early morning after an overnight fast with subjects abstaining from vigorous exercise for ≥24 h. Assessment of body composition was performed with a gold standard method viz. dual x-ray absorptiometry.

However, some limitations must be addressed. First of all, the control participants do not reflect the typical, less physically active Swiss population. For example, the METmax of the control participants was “only” two units less when compared to their endurance-trained counterparts. In addition, the fat mass percentage might be lower in the control participants compared to the general population. On the other hand, the physical activity level, body composition, and the aerobic capacity of the control participants might reflect the recommended “normal” human phenotype, i.e., physically active and “normal” BMI. Nevertheless, inclusion of a sedentary overweight (or obese) control group would give additional insights about the error when using the conventional 1-MET value for both determining the aerobic metabolic capacity and estimating the energy expenditure in these populations. Generally, it must be stated that estimating the energy expenditure using published MET values of the Compendium of Physical Activities must be taken with caution since the published MET values are often based on only one reference. Therefore, it can be expected that there is a wider variance in estimated AEE compared to the direct assessment of oxygen consumption during various physical activities [9]. In the present study, the dietary intake data (diary) were not analyzed, since the validity of self-reported energy intake data is highly questionable [72]. However, assessment of the total energy expenditure by use of objective validated tools (e.g., doubly labelled water) would have given further information about the interplay with energy requirements and aerobic metabolic capacity.

## 5. Conclusions

The use of a conventional 1-MET value appears inappropriate for determining the aerobic metabolic capacity and estimating the daily energy expenditure in active and endurance-trained persons. When the conventional standard 1-MET value was used, the predicted resting energy expenditure was slightly but significantly underestimated (above all in men). As a result, the calculation of METmax was significantly overestimated due to the underestimation of the denominator. Furthermore, the energy costs of non-maximal physical activities should also be underestimated when the conventional 1-MET value is used and this might lead to an underestimation of energy requirements for a given physical activity. For valid assessment of METmax (calculated from VO_2_max), measuring RMR by indirect calorimetry is recommended or, if not possible, estimating RMR is recommended using published validated equations tailored to the characteristics of the group studied in terms of age, gender, body composition (FFM), physiological status (i.e., pregnancy), and ethnicity. For estimating energy requirements, it can be recommended to (1) either measure directly the energy expenditure by use of validated tools, or (2) measure (or at least estimate) RMR and use appropriately adjusted MET values published in the literature [9] for estimating the energy costs of various structured exercises as well as free-living daily physical activities.

## Figures and Tables

**Figure 1 nutrients-11-00458-f001:**
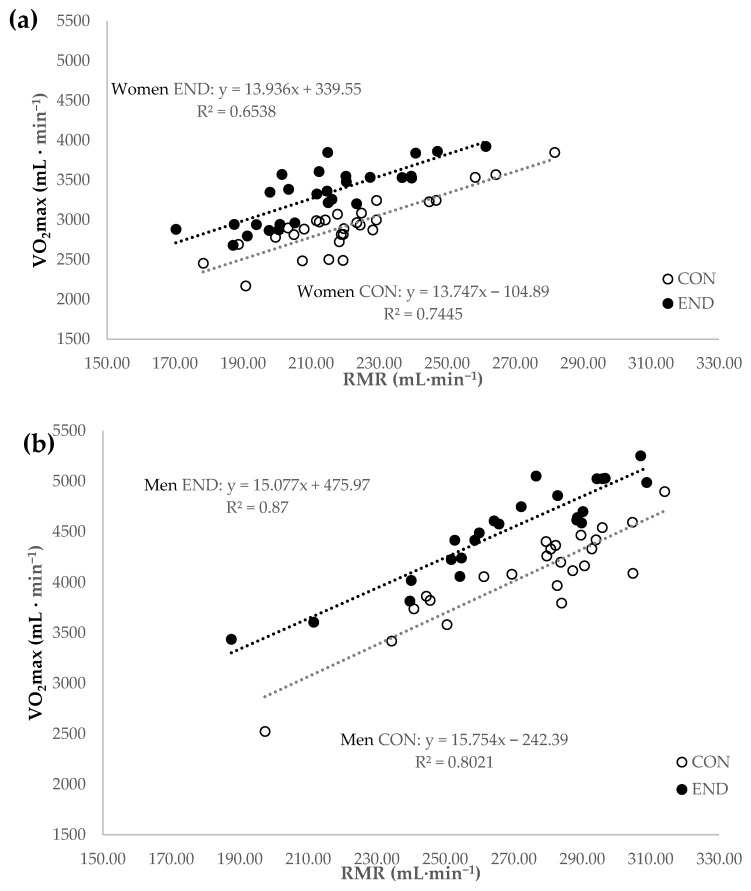
Resting metabolic rate (RMR) and maximum oxygen consumption (VO_2_max) in (**a**) women and (**b**) men who were endurance trained subjects (END) and healthy controls (CON).

**Figure 2 nutrients-11-00458-f002:**
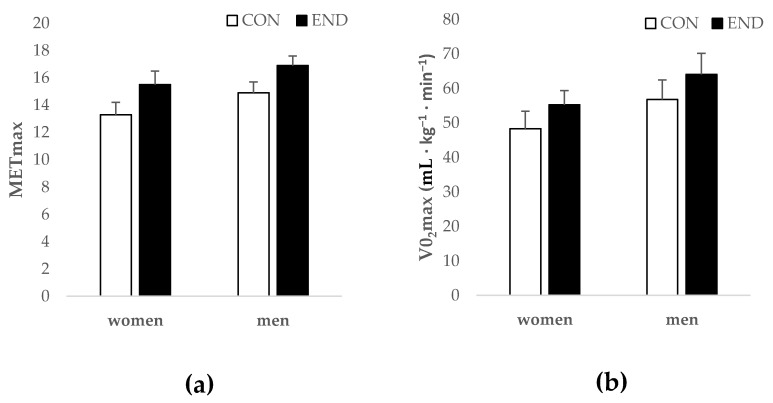
(**a**) Maximum metabolic equivalent of task (METmax) and (**b**) maximum oxygen consumption (VO_2_max) in women and men who were endurance trained subjects (END) and healthy controls (CON).

**Table 1 nutrients-11-00458-t001:** Overview about included endurance trained participants (END) and healthy controls (CON) with a valid resting metabolic rate (RMR) and maximum oxygen consumption (VO_2_max) measurements. Data are presented as Mean ± SD.

	Women		Men	
	CON (*n* = 29)	END (*n* = 28)	CON (*n* = 24)	END (*n* = 24)
Age (years)	27.6 ± 4.1	39.0 ± 6.1	27.0 ± 5.2	26.6 ± 5.0
Body mass (kg)	60.7 ± 6.7	59.6 ± 6.3	72.0 ± 7.4	70.8 ± 7.3
Height (cm)	167 ± 6	169 ± 6	178 ± 6	180 ± 6
BMI (kg∙m^−2^)	21.7 ± 1.6	20.8 ± 1.5 ^2^	22.7 ± 2.2	21.8 ± 1.8
Fat mass (%)	27.1 ± 5.5	23.7 ± 4.4 ^2^	15.9 ± 5.5	15.2 ± 4.7
FFM (kg)	45.0 ± 4.9	46.2 ± 4.9	61.3 ± 5.7	60.8 ± 7.2
PAL ^1^	1.8 ± 0.2	2.1 ± 0.2 ^4^	1.8 ± 0.3	1.9 ± 0.3
RMR				
(kcal∙day^−1^)	1505 ± 155	1457 ± 148	1873 ± 186	1824 ± 198
(kcal∙kg^−1^·day^−1^)	24.9 ± 1.9	24.5 ± 2.1	26.1 ± 1.8	25.9 ± 2.5
(kcal∙kg^−1^∙h^−1^)	1.04 ± 0.08 ^5^	1.02 ± 0.09	1.09 ± 0.07 ^7^	1.08 ± 0.10 ^6^
RMRpred (kcal∙kg^−1^∙h^−1^)	0.98 ± 0.05 ^5^	0.99 ± 0.05	1.03 ± 0.05 ^5^	1.04 ± 0.04 ^7^
RQrest	0.76 ± 0.04	0.76 ± 0.04	0.76 ± 0.06	0.75 ± 0.04
VO_2_max				
(L·min^−1^)	2.9 ± 0.4	3.3 ± 0.4 ^3^	4.1 ± 0.5	4.5 ± 0.5 ^3^
(mL∙kg^−1^∙min^−1^)	48.3 ± 5.1	55.3 ± 4.1 ^4^	56.8 ± 5.7	64.1 ± 6.1 ^4^

BMI = body mass index, FFM = fat-free mass, PAL = physical activity level, RMRpred = RMR predicted by use of the Harris-Benedict equation, RQrest = respiratory quotient at rest. ^1^ Valid Actiheart data available for 46 females (22 CON and 24 END) and 35 males (16 CON and 19 END). ^2^ Significantly different from CON of the same sex group (*p* < 0.05). ^3^ Significantly different from CON of the same sex group (*p* < 0.01). ^4^ Significantly different from CON of the same sex group (*p* < 0.0001). ^5^ Significantly different from the value of 1 kcal∙kg^−1^∙h^−1^ (*p* < 0.05). ^6^ Significantly different from the value of 1 kcal∙kg^−1^∙h^−1^ (*p* < 0.01). ^7^ Significantly different from the value of 1 kcal∙kg^−1^∙h^−1^ (*p* < 0.0001).

**Table 2 nutrients-11-00458-t002:** Values and concurrent validity of the maximum metabolic equivalent of tasks (METmax) by use of the individual (METmax_ind) and conventional (METmax_fix, 3.5 mL∙kg^−1^∙min^−1^) 1-MET-value for calculating in endurance trained participants (END) and healthy controls (CON). Data are presented as Mean ± SD.

	Women		Men	
	CON (*n* = 29)	END (*n* = 28)	CON (*n* = 24)	END (*n* = 24)
METmax_ind	13.3 ± 0.9 ^2^	15.5 ± 1.0 ^1^	14.9 ± 0.8 ^3^	16.9 ± 0.7 ^1,2^
METmax_fix	13.8 ± 1.4	15.9 ± 1.2 ^1^	16.3 ± 1.6	18.3 ± 1.8 ^1^
*r* value	0.69 ^4^	0.24	0.78 ^4^	0.10
MAE	0.9 ± 0.8	1.0 ± 0.9	1.5 ± 0.9	1.9 ± 1.3
MAPE (%)	6.6 ± 6.2	6.8 ± 6.2	10.3 ± 5.9	11.3 ± 7.6
SEE	0.63	1.03	0.52	0.70

^1^ Significantly different from CON of the same sex group (*p* < 0.0001). ^2^ Significantly different from METmax_fix of the same sex and experimental group (*p* < 0.01). ^3^ Significantly different from METmax_fix of the same sex and experimental group (*p* < 0.0001). ^4^ Correlation significant at *p* < 0.0001.

**Table 3 nutrients-11-00458-t003:** Calculation of activity energy expenditure (AEE) for one hour of activity (either light, moderate, vigorous, and very vigorous) by multiplication of the individual (AEE_ind), conventional (AEE_fix), and predicted (AEE_pred) 1-MET value with published MET values of specific activities [9]. Data are presented as Mean ± SD.

	Women		Men	
	CON (*n* = 29)	END (*n* = 28)	CON (*n* = 24)	END (*n* = 24)
Light activity (e.g., sitting tasks, Code 11580, 1.5 METs)				
AEE_ind (kcal∙h^−1^)	94 ± 10	91 ± 9	117 ± 12	114 ± 12
AEE_fix (kcal∙h^−1^)	91 ± 10 ^1^	89 ± 10	108 ± 11 ^3^	106 ± 11 ^2^
Mean difference AEE_ind—AEE_fix (kcal∙h^−1^)	3 ± 7	2 ± 8	9 ± 8	8 ± 10
AEE_pred (kcal∙h^−1^)	89 ± 5 ^2^	88 ± 5 ^1^	110 ± 8 ^2^	110 ± 8 ^2^
Mean difference AEE_ind—AEE_pred (kcal∙h^−1^)	6 ± 7	3 ± 8	7 ± 8	4 ± 8
Moderate activity (e.g., organizing room, Code 05125, 4.8 METs)				
AEE_ind (kcal∙h^−1^)	301 ± 31	291 ± 30	375 ± 37	365 ± 40
AEE_fix (kcal∙h^−1^)	292 ± 32 ^1^	286 ± 30	346 ± 36 ^3^	340 ± 35 ^2^
Mean difference AEE_ind – AEE_fix (kcal∙h^−1^)	10 ± 23	5 ± 24	29 ± 26	25 ± 33
AEE_pred (kcal∙h^−1^)	283 ± 16 ^2^	281 ± 17 ^1^	353 ± 26 ^2^	352 ± 27 ^2^
Mean difference AEE_ind—AEE_pred (kcal∙h^−1^)	18 ± 24	11 ± 24	22 ± 24	13 ± 27
Vigorous activity (e.g., stair climbing, Code 17130, 8.0 METs)				
AEE_ind (kcal∙h^−1^)	502 ± 52	486 ± 49	625 ± 62	608 ± 66
AEE_fix (kcal∙h^−1^)	486 ± 54 ^1^	477 ± 51	576 ± 60 ^3^	566 ± 58 ^2^
Mean difference AEE_ind—AEE_fix (kcal∙h^−1^)	16 ± 38	9 ± 41	48 ± 43	42 ± 55
AEE_pred (kcal∙h^−1^)	472 ± 27 ^2^	468 ± 29 ^1^	589 ± 43 ^2^	587 ± 44 ^2^
Mean difference AEE_ind—AEE_pred (kcal∙h^−1^)	30 ± 39	18 ± 40	36 ± 41	21 ± 45
Very vigorous activity (e.g., running 11 mph, Code 12130, 16 METs)				
AEE_ind (kcal∙h^−1^)	1004 ± 103	971 ± 99	1249 ± 124	1216 ± 132
AEE_fix (kcal∙h^−1^)	972 ± 107 ^1^	953 ± 101	1152 ± 119 ^3^	1132 ± 117 ^2^
Mean difference AEE_ind—AEE_fix (kcal∙h^−1^)	32 ± 75	18 ± 81	97 ± 85	84 ± 110
AEE_pred (kcal∙h^−1^)	944 ± 54 ^2^	934 ± 57 ^1^	1177 ± 85 ^2^	1174 ± 89 ^2^
Mean difference AEE_ind—AEE_pred (kcal∙h^−1^)	59 ± 78	36 ± 80	72 ± 81	42 ± 89

^1^ Significantly different from AEE_ind of the same experimental and sex group (*p* < 0.05). ^2^ Significantly different from AEE_ind of the same experimental and sex group (*p* < 0.01). ^3^ Significantly different from AEE_ind of the same experimental and sex group (*p* < 0.0001).

**Table 4 nutrients-11-00458-t004:** Total energy expenditure (TEE) calculated with the individual (TEE_ind), conventional (TEE_fix), and predicted (TEE_pred) 1-MET value for a sedentary or light activity lifestyle (PAL 1.53), an active or moderately active lifestyle (PAL 1.76), and a vigorous or vigorous active lifestyle (PAL 2.25) [45].

	Women		Men	
	CON (*n* = 29)	END (*n* = 28)	CON (*n* = 24)	END (*n* = 24)
Sedentary or light activity lifestyle (PAL 1.53)				
TEE_ind (kcal∙day^−1^)	2303 ± 237	2229 ± 226	2866 ± 285	2790 ± 303
TEE_fix (kcal∙day^−1^)	2230 ± 246 ^1^	2188 ± 232	2645 ± 273 ^3^	2598 ± 268 ^2^
Mean difference TEE_ind—TEE_fix (kcal∙day^−1^)	73 ± 173	41 ± 186	222 ± 196	192 ± 253
TEE_pred (kcal∙day^−1^)	2167 ± 124 ^2^	2146 ± 131 ^1^	2702 ± 195 ^2^	2694 ± 204 ^2^
Mean difference TEE_ind—TEEpred (kcal∙day^−1^)	136 ± 180	83 ± 183	164 ± 186	96 ± 205
Active or moderately active lifestyle (PAL 1.76)				
TEE_ind (kcal∙day^−1^)	2649 ± 272	2564 ± 260	3297 ± 327	3210 ± 348
TEE_fix (kcal∙day^−1^)	2565 ± 283 ^1^	2517 ± 267	3042 ± 314 ^3^	2989 ± 308 ^2^
Mean difference TEE_ind—TEE_fix (kcal∙day^−1^)	84 ± 199	47 ± 214	255 ± 225	221 ± 291
TEE_pred (kcal∙day^−1^)	2493 ± 143 ^2^	2468 ± 151 ^1^	3108 ± 224 ^2^	3099 ± 234 ^2^
Mean difference TEE_ind—TEEpred (kcal∙day^−1^)	157 ± 207	95 ± 211	189 ± 215	111 ± 236
Vigorous or vigorous active lifestyle (PAL 2.25)				
TEE_ind (kcal∙day^−1^)	3387 ± 348	3277 ± 332	4215 ± 418	4102 ± 445
TEE_fix (kcal∙day^−1^)	3279 ± 361 ^1^	3217 ± 342	3889 ± 402 ^3^	3821 ± 394 ^2^
Mean difference TEE_ind—TEE_fix (kcal∙day^−1^)	108 ± 254	60 ± 273	326 ± 288	282 ± 372
TEE_pred (kcal∙day^−1^)	3186 ± 183 ^2^	3156 ± 193 ^1^	3974 ± 287 ^2^	3962 ± 300 ^2^
Mean difference TEE_ind—TEEpred (kcal∙day^−1^)	201 ± 265	122 ± 270	241 ± 274	141 ± 301

^1^ Significantly different from TEE_ind of the same experimental and sex group (*p* < 0.05). ^2^ Significantly different from TEE_ind of the same experimental and sex group (*p* < 0.01). ^3^ Significantly different from TEE_ind of the same experimental and sex group (*p* < 0.0001).

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
