# Peer review of "Comparison of Conventional and Individualized 1-MET Values for Expressing Maximum Aerobic Metabolic Rate and Habitual Activity Related Energy Expenditure"

_nutrients, 2019, doi:10.3390/nu11020458_

Reviewer 1 Report

This article reads well and is engaging. The introduction and methods are described clearly so to focus the article and explain the procedures. The discussion reads well. The strengths and limitations are important and identified well. The choice of the subjects limit external to the population as a whole, but this is noted in the article. The discussion and conclusion are written well and on target with the issues.  There are a few comments to the Tables/figures and the discussion that you should consider.

SPECIFIC COMMENTS

Figure 1 – please identify the sex within each figure next to the equation for each viewing. Perhaps add something like this (a = women) (b = men).

Section 4.1 discussion and line 377– as the Compendium values may have been based on only one study (see references for each activity), the results of comparing measured vs. estimated values with the Compendium must be taken with caution and expect wider variance.  Please make a note of this in the discussion.

Please include somewhere in the discussion or conclusion - the only true way to get precise EE of a day’s activities is to measure the oxygen cost of all activities performed for each individual or use DLW to measure (estimate really) daily kcals expended.

The end of section 4.3 shows the complexity of estimating energy cost of activities by sex, method of correction of REE or use of standard METs.  Good comments made about this complexity.

Author Response

SPECIFIC COMMENTS

Figure 1 – please identify the sex within each figure next to the equation for each viewing. Perhaps add something like this (a = women) (b = men).

Response: We added the sex next to the equations in Figure 1.

Section 4.1 discussion and line 377– as the Compendium values may have been based on only one study (see references for each activity), the results of comparing measured vs. estimated values with the Compendium must be taken with caution and expect wider variance.  Please make a note of this in the discussion.

Response: We added a paragraph stating that estimation of energy expenditure by use of the Compendium values must be taken with caution, since the underlying MET values are often based by only one study (lines 434-438).

Please include somewhere in the discussion or conclusion - the only true way to get precise EE of a day’s activities is to measure the oxygen cost of all activities performed for each individual or use DLW to measure (estimate really) daily kcals expended.

Response: We added a sentence in the discussion where it is stated that neither the use of a corrected or standard 1-MET value can replace the direct assessment of energy expenditure, by using oxygen consumption or DLW (lines 407-409).

The end of section 4.3 shows the complexity of estimating energy cost of activities by sex, method of correction of REE or use of standard METs.  Good comments made about this complexity.

Response: Thank you!

Reviewer 2 Report

The manuscript is well presented and written and present interesting findings. However, some minor modifications are needed:

1-      Please add some hypothesis at the end of the Introduction.

2-      I suggest changing “We” and “our” in all parts of the manuscript.

Author Response

The manuscript is well presented and written and present interesting findings. However, some minor modifications are needed:

1-      Please add some hypothesis at the end of the Introduction.

Response: We added a hypothesis at the end of the introduction (line 80-82).

2-      I suggest changing “We” and “our” in all parts of the manuscript.

Response: We changed „we“ and „our“ in the entire manuscript.

Reviewer 3 Report

It is an interesting manuscript, with a relevant subject. However, it could be of interest if authors better present the "study problem/relevance" in the introduction and also better describe the statistical methods used, as well as the variables that did not present a normal distribution

Author Response

It is an interesting manuscript, with a relevant subject. However, it could be of interest if authors better present the "study problem/relevance" in the introduction and also better describe the statistical methods used, as well as the variables that did not present a normal distribution

Response: To address better the study problem, we added a paragraph that mis-estimation of energy requirements can promote a negative or positive energy balance and thus unexpected and undesirable weight fluctuations (lines 68-70), which is relevant not only for athletes but also for individuals from the general population. We critically reviewed our section on the statistical methods. The way any not-normal distribution was accounted for is clearly indicated.